# Model-based identification of conditionally-essential genes from transposon-insertion sequencing data

Vishal Sarsani[1], Berent Aldikacti[2], Shai He[1], Rilee Zeinert[3], Peter Chien[2], Patrick Flaherty[1]*

**1** Department of Mathematics and Statistics, University of Massachusetts Amherst, Amherst, Massachusetts, United States of America, **2** Department of Biochemistry and Molecular Biology, University of Massachusetts Amherst, Amherst, Massachusetts, United States of America, **3** Division of Molecular and Cellular Biology, Eunice Kennedy Shriver National Institute of Child Health and Human Development, Bethesda, Maryland, United States of America

* pflaherty@umass.edu

## Abstract

The understanding of bacterial gene function has been greatly enhanced by recent advancements in the deep sequencing of microbial genomes. Transposon insertion sequencing methods combines next-generation sequencing techniques with transposon mutagenesis for the exploration of the essentiality of genes under different environmental conditions. We propose a model-based method that uses regularized negative binomial regression to estimate the change in transposon insertions attributable to gene-environment changes in this genetic interaction study without transformations or uniform normalization. An empirical Bayes model for estimating the local false discovery rate combines unique and total count information to test for genes that show a statistically significant change in trans-poson counts. When applied to RB-TnSeq (randomized barcode transposon sequencing) and Tn-seq (transposon sequencing) libraries made in strains of *Caulobacter crescentus* using both total and unique count data the model was able to identify a set of conditionally beneficial or conditionally detrimental genes for each target condition that shed light on their functions and roles during various stress conditions.

## Author summary

Transposon insertion sequencing allows the study of bacterial gene function by combining next-generation sequencing techniques with transposon mutagenesis under different genetic and environmental perturbations. Our proposed regularized negative binomial regression method improves the quality of analysis of this data.

This is a *PLOS Computational Biology* Methods paper.

**Data Availability Statement:** The code used in this part is part of the R package and is freely available at https://github.com/vsarsani/rnbtn. The data files are freely available at https://doi.org/10.6084/m9.figshare.17136821.v1.

**Funding:** PF is supported by National Institutes of Health R01 GM135931 (www.nih.gov) and National Science Foundation HDR TRIPODS 1934846 (www.nsf.gov). PC is supported by NIH grant R35GM130320 (www.nih.gov). The funders had no role in study design, data collection and analysis, decision to publish, or preparation of the manuscript.

**Competing interests:** The authors have declared that no competing interests exist.

## Introduction

A central question in molecular genetics is, What genes are essential for life? Prior to the advent of high-throughput technology this question was addressed by mutagenesis and fine mapping [1, 2]. The simplicity of homologous recombination in *S. cerevisiae* allowed for the generation of a complete mutant library containing strains each with a complete knockout of a single gene and tagged with a unique genetic barcode[3]. Subsequent analysis of this library by custom microarrays and sequencing revealed genes essential for growth in rich media as well as *conditionally essential* genes—genes that are dispensable in rich media, but are essential in different environmental conditions [3–5]. However, generating a mutant pool from individual genetic knockout strains is labor-intensive and not feasible in organisms for which homologous recombination is inefficient. Transposon sequencing (Tn-seq) methods have alleviated this problem and provide a powerful method for identifying essential and dispensable genes under a variety of environmental conditions and genetic backgrounds. This type of study, with perturbations to both the genetic content and the environmental context (gene × environment) is typically referred to as a genetic interaction study; here the primary phenotype is growth.

### Transposon sequencing

Transposon sequencing uses a modified transposon to generate a saturation mutant library of a background strain of interest. Each transposon has a selectable marker; a unique, random DNA barcode (in some cases); and loci for PCR amplification that can be used to identify the DNA adjacent to the transposon insertion site [6, 7]. Once the transposon mutant library is generated, it can be grown in various environmental conditions of interest. Strains that have a fitness defect due to the transposon insertion grow more slowly or not at all. The abundance of the transposon insertion mutant strain in the library can be assayed by sequencing the library after growth and counting the reads that map to a particular insertion site. For each gene, the change in the count of sequenced transposon insertions between the control and the perturbed environment can be used to identify conditionally essential or conditionally dispensable genes.

Since the introduction of the original Tn-seq method, many variations have been developed to facilitate the study of a wider range of organisms or to improve efficiency [6]. Random-barcode transposon sequencing amortizes the cost of multiple environmental perturbation experiments by doing the expensive mapping of transposon insertion site to random barcodes once and then using that mapping for all future experiments [8]. Transposon sequencing technology addresses the time consuming and often technically challenging process of generating one-at-a-time gene deletions by using parallel mutagenesis and counting-by-sequencing[9]. But, this technology has introduced a new, statistical problem. How can the transposon count data be used to test the hypothesis that a gene is essential such that all of, and only, the essential and dispensable genes are identified?

### Related work

There are several existing statistical approaches for analyzing transposon sequencing data. van Opijnen et al. [10] used several normalization steps to compute a ratio of the fold-expansion of the mutant relative to the rest of the population. Then, a t-test with a Bonferroni correction was used for each gene to decide if a change in the fitness statistic is significant. This type of normalization renders the statistic independent of growth duration, but requires an additional calibration experiment to estimate an expansion factor which measures the growth of the bacterial population during library selection. The fitness effect estimator is non-linearly

dependent on the calibration factor because it appears in both a logarithm and in the denominator of the fitness effect ratio.

Wetmore et. al. [8] dispensed with the calibration step and still found good estimates of fitness effect. They computed the log-ratio of start-time $t_0$ count to the stop-time $t_{after}$ count. They added a pseudo-count term to regularize noisy estimates for low counts. These low count observations were filtered out in [10].

ESSENTIALS is a software package developed by [11] that uses Loess [12] normalization followed by the application of edgeR [13], a software package developed for identifying differentially expressed genes from RNA-seq data, to call essential genes. They demonstrated that their package is robust to differences in transposon sequencing technology—a significant benefit as TnSeq experimental methods continue to be revised and improved.

DeJesus et. al. [14] developed a full Bayesian model for Tn-seq count data. They approached the problem by defining a Boolean variable to represent whether a gene is essential or nonessential. In their method, the data for a gene includes the number of insertions, the longest run of non-insertions, and the span of nucleotides of the longest run of non-insertions. This additional information beyond the number of insertion counts is informative and the Bayesian model elegantly incorporates all of the data into a posterior probability of essentiality. In other work, DeJesus et. al. [15] identified genetic interactions by measuring changes in enrichment through a hierarchical Bayesian model. Their method identified mutants that differentially affect bacterial fitness by performing a four-way comparison. While this is primarily useful for small-scale genetic interaction studies, it cannot accommodate multiple varying conditions and stress levels nested within many background strains. The normal distribution used in the paper may not properly approximate the distribution of insertions, especially for small genes compared to a negative binomial model.

Subramaniyam et. al. [16] focuses on fine-resolution mapping of essential regions. Their method applies to transposon libraries constructed with the *mariner* transposon family which preferentially inserts in TA dinucleotides. Their method models the number of transposon insertions at each TA dinucleotide site rather than aggregating by gene. Because many TA dinucleotide sites are unlikely to harbor any transposon insertions, they employ a zero-inflated negative binomial model to accommodate the many zero counts. It should be noted that [10] and [8] include a normalization for the number of Tn counts at the start of the experiment, but more recent model-based work does not require this normalization [14, 16].

## Contributions

Our work builds upon these previous works in several ways. Like [16], our approach employs a negative binomial generalized linear model to use information from the entire experimental data set rather than using only pairs of experiments. Our model employs a Bayesian prior over coefficients as in [14] that manifests as a regularization term in the regression formulation. Our work differs from these efforts in that we aggregate transposon counts at the gene level in the context of a negative binomial model with nested effects which allows our model to be robust to the transposon library creation method, and we use a false discovery rate approach to call conditionally beneficial and conditionally detrimental genes. Our contributions are: (1) a regularized negative binomial model with nested effects to estimate the effect of varying environmental conditions in the context of genetic background, (2) the use of both unique Tn insertions and total Tn insertions to improve sensitivity and specificity, (3) the use of an intersection local false discovery rate control to identify genes that have a quantitative effect on growth.

## Total and unique insertions

Total counts are obtained by summing up the counts across all the insertion sites in a gene. Total counts tend to reflect the growth of bacteria under a particular condition, which is extremely useful for measuring overall fitness effects. While total counts are a useful measure to identify conditionally beneficial and conditionally detrimental genes, they may be skewed when there is an inconsequential insertion hotspot near the tail ends due to an amplification error or a genome-specific bias. Unique counts are obtained by counting the number of insertion sites in a gene. Unique counts capture site-specific variations and satisfy independent assumptions of the count models. Unique counts provide a deeper understanding of genomic location-specific fitness effects. But, the sparse nature of these counts and library dependence makes it somewhat unreliable for modeling.

## Problem statement

Let the set of genes under investigation be $\mathcal{S}$. We partition $\mathcal{S}$ into a set of essential genes, $\mathcal{A}$, and the set of nonessential genes, $\overline{\mathcal{A}} = \mathcal{S} \setminus \mathcal{A}$. Essential genes show no growth when disrupted in a wild-type strain in standard media. Then, we partition the set of nonessential genes into a set of conditionally essential genes, $\mathcal{B} \subseteq \overline{\mathcal{A}}$, and a set of conditionally nonessential genes, $\overline{\mathcal{B}} = \overline{\mathcal{A}} \setminus \mathcal{B}$. The conditionally essential set contains genes that when disrupted still yield growth in control conditions, but do not grow when the genetic background or media conditions are varied; this set depends on the growth condition.

In order to identify genes that produce a quantitative, rather than qualitative, change in fitness as measured by growth rate, the set of conditionally nonessential genes is further partitioned into the following mutually exclusive, collectively exhaustive sets: conditionally beneficial ($\mathcal{C}$), conditionally neutral ($\mathcal{D}$), and conditionally detrimental ($\mathcal{F}$). If the disruption of a gene decreases growth, it is called conditionally beneficial ($\mathcal{C}$). If the disruption of a gene does not affect growth, it is called conditionally neutral ($\mathcal{D}$). And, if the disruption of a gene improves growth, it is called conditionally detrimental ($\mathcal{F}$).

Denote the genetic background of the experiment $g \in \mathcal{G}$ and the environmental condition $e \in \mathcal{E}$. Note that not all pairwise combinations in $\mathcal{G} \times \mathcal{E}$ may be available in a data set. For a given combination $(g, e)$ the data set contains $R_{ge}$ replicate experiments; we index the replicate with $r$. In experiment $(g, e, r)$, there are $N_{ger}$ observed transposon insertions that are mapped to genes (perhaps excluding some trimmed region around the start and stop codon of the gene). We reduce the raw data to two features for each gene: (1) the **total** count of insertions and (2) the count of **unique** insertions. For gene $i$ and experiment $(g, e, r)$, let $y_{geri}^{\text{tot}}$ be the total count of insertions and let $y_{geri}^{\text{uniq}}$ be the count of unique insertions. The average total counts across $r$ replicates is $\bar{y}_{gei}^{\text{tot}}$, and the average unique counts is similarly defined $\bar{y}_{gei}^{\text{uniq}}$. We define the control condition to be a wild-type genetic background within standard PYE media and we denote this condition $(g', e')$. The average control condition counts for gene $i$ are then $\bar{y}_{g'e'i}^{\text{tot}}$.

Previous work has focused primarily on the estimation of the set of genes that produce a qualitative elimination of growth when disrupted. In this work, we estimate the set of *essential genes* using only the total counts as $\widehat{\mathcal{A}} = \left\{ i | \bar{y}_{g'e'i}^{\text{tot}} < 1, \forall i \in \mathcal{S} \right\}$, noting that $\bar{y}_{g'e'i}^{\text{tot}} \geq \bar{y}_{g'e'i}^{\text{uniq}}$. We estimate the set of *conditionally essential genes* in condition $(g, e)$ as $\widehat{\mathcal{B}}_{ge} = \left\{ i | \bar{y}_{gei}^{\text{tot}} < 1, \forall i \in \overline{\mathcal{A}} \right\}$. A hierarchical decision tree representation of these categories is shown in Fig 1.

The goal of this work is to identify the set of genes that have a quantitative effect on the ability of the organism to grow under certain genetic and environmental conditions. The null hypothesis ($H_0$) is that a gene is conditionally neutral. To identify the set $\mathcal{R} = \mathcal{C} \cup \mathcal{F}$ of conditionally

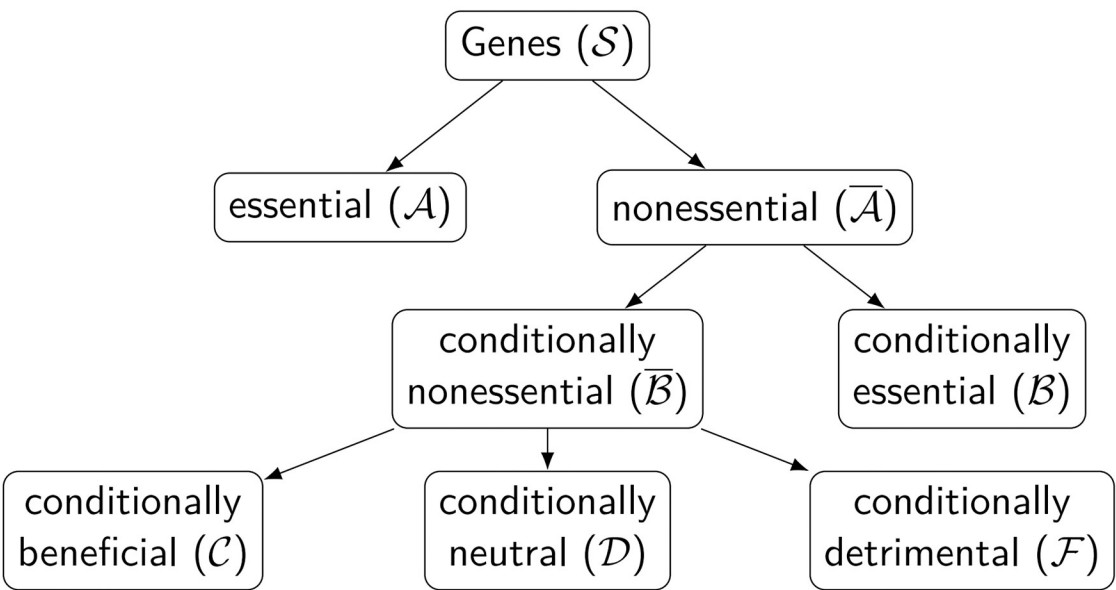

**Fig 1. Gene categorization in transposon insertion sequencing (Tn-seq).** Our modeling framework primarily focuses on identifying the set of conditionally beneficial $\mathcal{C}$ (decreased growth due to disruption) and conditionally detrimental genes $\mathcal{F}$ (increased growth due to disruption).

beneficial and conditionally detrimental genes is often too challenging and instead a more approachable task is to ensure that the rate of false discoveries in $\mathcal{R}$ is bounding in probability. Therefore, the problem is to estimate the set $\mathcal{R}$ such that the false discovery rate is small.

## Materials and methods

### RB-TnSeq experimental methods

RB-TnSeq uses a randomly barcoded transposon to amortize the cost of many related experiments [8]. Barcoded transposon donor plasmids are transferred to the cell of interest by either electroporation or conjugation. Subsequently, cells containing plasmids are selected using a selection media, and small aliquots are frozen in 10% glycerol. The frozen aliquot is the mutagenesis libraries used in all experiments. In RB-Tnseq, a sequencing run is done on the libraries to assign each barcode to its genomic location. For subsequent experiments on these libraries, a simple single PCR step is required to amplify and count the barcodes.

 **Read mapping and pre-processing.** In this study, we used RB-TnSeq data of *Caulobacter crescentus* and *Pseudomonas fluorescens FW300-N1B4* from Price et. al. [17]. As input we have downloaded `all.poolcounts` (http://genomics.lbl.gov/supplemental/bigfit/), and generated two different count files from it. The first, labeled "total counts", are the sum of all insertions aggregated by each gene. The second is the "unique counts", where instead of using the sum of all insertions, we have used the sum of the number of unique barcodes that have non-zero reads per gene.

### Tn-seq experimental methods

Transposon mutagenesis libraries used in this study were generated as previously described [18]. Briefly, wild-type (wt) and Δ*lon Caulobacter crescentus NA1000* strains were grown until mid-log phase, pelleted, washed three times with 10% glycerol, and transformed with EzTn5 <Kan-2> transposomes (Lucigen) by electroporation. Following recovery in PYE,

transformed cells were plated on PYE + Kan selection media and grown for 7 days. Colonies were scraped, pooled, and frozen in PYE + 20% glycerol in 1 ml aliquots and frozen for further experiments. For stress condition experiments, 2 aliquots of each library was thawed and separately recovered overnight in 2 x 10 ml of PYE in a 30˚C shaker. These saturated cultures were then stressed as described below. All conditions were performed in quadruplicates, optical density (OD) measurements were taken at 600 nm.

**Control environment.**   Libraries were back diluted to OD 0.008 into 7 ml of PYE and grown overnight until they reach saturation at OD $\sim 1.6$.

**Heat shock stress.**   One ml of the overnight culture was heat-shocked at 42˚C for 45 minutes in a heat-block, then back-diluted to OD 0.008 and grown overnight until saturation.

**L-canavanine.**   Overnight cultures of cells were back diluted to OD 0.008 in 7 ml of PYE + 100 ug/ml L-canavanine and grown at 30˚C for 90 minutes. After 90 minutes of L-canavanine stress, the cells were spun for 10 minutes at 5000 rpm, washed once with PYE, spun again, then resuspended with 7 ml of PYE, and recovered overnight until they reached saturation.

**Library preparation.**   Following overnight growth, 1.5 ml of saturated culture from each Tn library was pelleted at 15,000 RPM for 1 minute and gDNA was extracted by MasterPure Complete DNA and RNA purification kit according to manufacturer's protocol. Sequencing libraries were prepared for Next-generation sequencing via three PCR steps. Indexed libraries were pooled and sequenced at the University of Massachusetts Amherst Genomics Core Facility on a NextSeq 500 (Illumina).

**Read mapping and pre-processing.**   Mapping and pre-processing of the Tnseq raw data was done as described previously with some modifications [18]. Briefly, samples were de-multiplexed, and unique molecular identifiers (UMIs) were added during PCR steps removed using Je [19]. Clipped reads mapped to the *Caulobacter crescentus* NA1000 genome (NCBI Reference Sequence: NC011916.1) using bwa, sorted with samtools [20, 21]. Duplicate transposon reads removed by Je and indexed with samtools. Genome positions are assigned to the 5′ position of transposon insertions using bedtools genomecov [22]. Subsequently, the bedtools map used to count either the total number of transposon insertions per gene using the bedtools map -o sum argument or the unique number of insertions using the bedtools map -o count argument.

**In-vivo validation.**   Overnight cultures of wild-type and Δ*clpA Caulobacter crescentus* strains each mixed at a 1:1 ratio with a reporter strain constitutively expressing fluorescent *Venus* (CPC798). The mixtures were kept at either 30˚C or heat-shocked at 42˚C for 45 minutes in a thermocycler. After the heat-shock, the mixtures were diluted to 1:4000 in PYE media and allowed to grow for 24 hours ($\sim 12$ doublings) at 30˚C. Number of fluorescent control (Venus) and nonfluorescent tester (WT or Δ*clpA*) cells were counted in both the initial mixture and after 24 hour growth using phase contrast and fluorescent microscopy. The same tester and control normalization coefficients were used for initial and 24 hour time points for each strain (normalization coefficient = 1/(tester and control) at time = 0). and time = 24 by adjusting the time = 0 ratios to 1 for each strain. Normalized 24 hour ratios are what we are reporting as competitive index. An index of greater than one means the tester condition were able to grow faster compared to the control and an index of less than one means the tester grew slower compared to the control. Quantifications of at least 100 cells were performed for each condition with replicates when possible.

## Regularized negative binomial regression

Our approach for integrating all of the experimental data to estimate the effect of the genetic background and the environmental condition is based on a generalized linear model

framework. Here, we describe the negative binomial model framework, the nested effects model matrix structure, and the form and rationale for regularization.

**Negative binomial model.**   The generalized linear model consists of three components: (1) a probability distribution for the sampling error, (2) a model matrix structure, and (3) a link function connecting the expected value of the response to the covariates. It has been observed that Tn-seq count data is often overdispersed and therefore, the data is better fit by a negative binomial distribution rather than a Poisson distribution because of the additional free parameter to allow for a variance that does not directly depend on the mean parameter. The link function that is often chosen for a negative binomial distribution is a log function and we do so here. The generalized linear model takes the form $E(\mathbf{y}_i|\mathbf{x}) = f^{-1}(\mathbf{x}\beta)$, where $\mathbf{y}_i$ is the vector of observed Tn counts across all experiments in the data set for gene $i$, $\mathbf{x}$ is the model matrix, $\beta$ is the vector of parameters, and $f^{-1}$ is the log link function.

**Nested effects in generalized linear regression model.**   The model matrix must be designed to specifically address the questions of interest of the data. First, we are interested in the main effect of the genetic background in relation to the wild-type strain. For example, if a there is a drastic reduction in Tn counts in a mutant background relative to wild-type, it indicates that the gene is beneficial conditional on the strain mutation(s). Likewise if there is a drastic increase in Tn counts in a mutant background relative to wild-type, the gene is likely detrimental conditional on the strain mutation(s). Second, we are interested in the effect of the environmental condition, but only in the context of the genetic background. For example, if there is a reduction in Tn counts in the $g = \Delta lon$ background relative to the wild-type background in rich media growth conditions, but then no change when shifted to a heat-stress, the gene may be viewed as interesting in the genetic background, but not in the conditions specific to heat-stress. One would expect that if a gene is beneficial in the $g = \Delta lon$ background that it continues to be beneficial in all environmental conditions—only deviations from that expectation should be flagged as scientifically interesting. These questions of interest logically lead to the consideration of a nested effects model matrix structure:

$$E(\mathbf{y}_i|\mathbf{x}) = f^{-1}(\beta_0 + \mathbf{x}_g\beta_g + \mathbf{x}_{e|g}\beta_{e|g}), \tag{1}$$

where $\mathbf{x}_g$ and $\mathbf{x}_{e|g}$ are the standard indicator matrix encodings for the genetic background and nested environmental condition respectively. Note that this nested model matrix structure is different than the one usually employed for modeling interactions in that there is no term corresponding to the main effects of the environmental condition $\mathbf{x}_e$. Structuring the model matrix in this way allows the inferential products of the model (the model parameters) to inform the scientifically interesting questions we have of the data. Fig 2 shows a hierarchical diagram of the nested effects interrogated by the generalized linear regression model.

A way to interrogate this data is to observe the baseline number of total and unique insertions in the wild-type background strain with no stress (control). An excess or depletion of insertions in the $\Delta lon$ background are viewed as a shift from the control. Finally, an excess or depletion of the stress conditions is viewed relative to the particular background strain the library was created in. This interpretation of the data leads to the nested effects model proposed here.

**Regularization.**   Estimating the model parameters when the number of transposon count is small has been noted by others and handled either by filtration [10] or the addition of pseudo-counts [8]. The low counts in response variables can result in inflated regression coefficients and are susceptible to very high variance. They also affect false discovery rate procedures increasing the risk of type-I errors. For genes that are conditionally nonessential ($\overline{\mathcal{B}}$), we employ a regularization methodology that has proven successful in many statistical contexts and has Bayesian as well as classical statistical rationale [23–25]. Regularization can be viewed

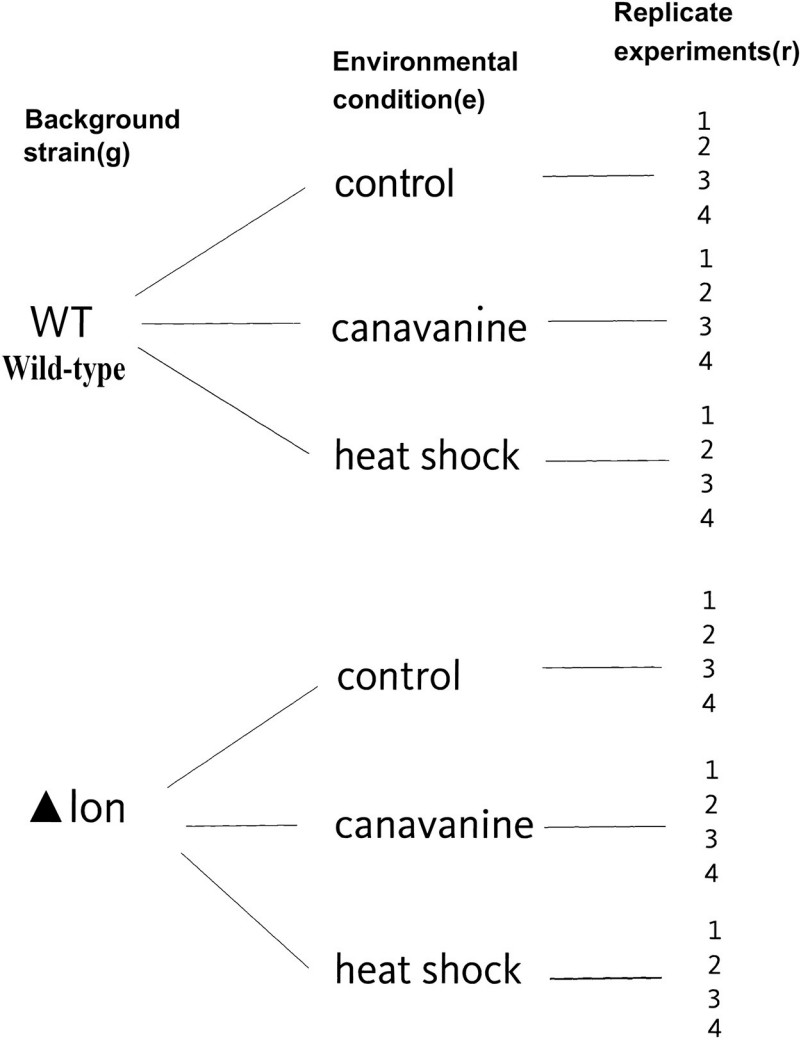

**Fig 2. Example of nested experimental design of Tn-seq data.** Shown are two background strains: WT and Δ*lon*, and three nested environmental perturbations: control, canavanine, heat shock. Each perturbation experiment is replicated four times.

as a prior distribution on the regression coefficients,

$$\beta \sim \text{Gaussian}(\lambda). \tag{2}$$

The Gaussian prior converts the maximum likelihood estimation problem for the regression coefficients to a penalized maximum likelihood estimation problem with an $L_2$ norm penalty or equivalently a maximum a-posteriori estimation problem. The parameters for the penalized count regression are estimated by a combination of the iteratively reweighted least squares (IRLS) algorithm and coordinate descent algorithm as implemented in the `mpath` package [26].

We have found that this regularization effectively shrinks large coefficient estimates due to small Tn counts. However, it does not address situations where there are exactly zero counts. In those cases, our model is not necessary—the gene can be considered conditionally essential in the condition with high confidence. Therefore, we restrict our modeling to conditionally nonessential genes ($\overline{\mathcal{B}}$).

### Local false discovery rate

The regularized negative binomial generalized linear model was fit to both the total count data, $\mathbf{y}_i^{\text{tot}}$, and the unique count data, $\mathbf{y}_i^{\text{uniq}}$ independently for each gene $i$. The next task is to decide if a gene is conditionally beneficial or conditionally detrimental or neutral. In a generalized linear model the response is conditionally independent of a covariate given all the other covariates in the model if and only if the associated model coefficient is equal to zero (for proof see [27]). Therefore, under the model-based framework testing if a gene is conditionally beneficial or detrimental is equivalent to testing whether the model coefficient is equal to zero.

Under the assumption that a large fraction of the genes under investigation are neutral, the local false discovery rate can be used to control the proportion of false positives in the set of called conditionally beneficial or conditionally detrimental genes [28]. The central idea is to fit a Gaussian distribution to the center of the empirical distribution of coefficients for a given effect across all genes. Genes that have a coefficient that is unlikely under that distribution are called conditionally beneficial or conditionally detrimental. There is abundant theory to support the use of this procedure to control the proportion of false discoveries [29–31].

The false discovery rate of the regression coefficient is

$$\text{Fdr}(\beta_{ic}) = \text{Prob}\{\text{gene } i \text{ is null in condition } c \mid |\beta_{ic}| \geq \bar{\beta}\} \tag{3}$$

The local false discovery rate makes use of a mixture model framework with two components. It fits a Gaussian distribution to the center of the empirical distribution of the regression coefficients $\beta_{ic}$ across genes. Genes associated with coefficients that are not attributable to the central Gaussian are called conditionally beneficial or conditionally detrimental [32].

**Intersection of marginal local false discovery tests.**   The standard false discovery rate approach only considers the coefficients estimated from one model, however, in our analysis, we estimate coefficients from the model fit to $\mathbf{y}^{\text{tot}}$ and the model fit to $\mathbf{y}^{\text{uniq}}$. Yet, we would like a single decision as to whether the gene is conditionally neutral or not. Our approach is to take the intersection of the decisions from the two models. That is, only genes that are deemed conditionally beneficial or conditionally detrimental on the basis of both unique counts and total counts are retained. This approach has the effect of reducing the number of calls and thus the number of false positives at the expense of false negatives.

## Results

We generated simulated data on 4,000 genes under 3 simulated knockout backgrounds and 4 environmental conditions with 5 replicates for each combination of strain background and environment. We compared the fit of the regularized negative binomial model to a zero-inflated negative binomial model of the type used by [16] and to a unregularized negative binomial model [11].

Our method was then applied to two independent data sets using different transposon sequencing methods. First, our method was applied to RB-TnSeq data. This data set explored the essential genes in many organisms across varying carbon sources, nitrogen sources, and environmental stress conditions. We selected only the *Caulobacter crescentus* data set for this study. The background genotype for all the RB-TnSeq experiments is wild-type so no synthetic lethality combinations are identifiable. Second, our method was applied to Tn-seq data that was collected in our lab. Both wild-type and a Δlon knockout strain were used as genetic backgrounds for library preparation. These strain pools were subjected to heat-shock stress and canavanine. Each condition was replicated at least two times in biological replicates.

## Simulation experiments

We simulated samples from total of three background strains ($g$) with four conditions ($e$) and each condition having five replicates ($r$). First the dispersion parameter was sampled from a Gamma distribution for each condition and for 8 intervals ($l$) each containing 500 genes. The hyper-parameters of the Gamma distribution were drawn from uniform distributions as

$$
\begin{aligned}
a_g &\sim U(0,5), \ \ b_g \sim U(0,5) \quad \text{for} \quad g = 1, 2, 3, \\
\theta_{gel} &\sim \text{Gamma}(a_g, b_g) \qquad \text{for} \quad g = 1, 2, 3, e = 1, \ldots, 4, l = 1, \ldots, 8.
\end{aligned}
\tag{4}
$$

The number of unique insertions for each gene was sampled from a negative binomial distribution with mean parameters shared across groups of 500 genes, $\mu = (0.5, 1, 2, 4, 8, 16, 32, 64)$,

$$
\mathbf{y}_{gerl}^{\text{uniq}} \sim NB(\mu_l, \theta_{sl}).
\tag{5}
$$

This simulation provides the number of *unique* transposon counts for each gene. For every gene, the total transposon insertion counts were obtained by sampling from a negative binomial distribution with mean $\mu = 100$ and dispersion $\theta = 1$ for each unique insertion site previously generated

$$
y_{geri}^{\text{tot}} \sim \sum_{s=1}^{y_{geri}^{\text{uniq}}} NB(\mu = 100, \theta = 1).
\tag{6}
$$

**Regularized negative binomial model reduces over-fitting.**   Out of 4,000 simulated genes, there were 82 for which the regularized negative binomial model fitting algorithm did not converge leaving 3,918 simulated genes for comparison to other algorithms. We observed that for 3,456(86.67%) genes, the regularized negative binomial model had a better fit as measured by residual variance compared to a unregularized negative binomial model [11]. Fig 3 shows the mean counts and the residual variance for each of the 3,918 genes for a multi-condition setup. Clearly, the negative binomial model alone fits poorly for low mean count values. S1 Fig shows the mean counts and the residual variance for a control condition setup. Even though the regularized negative binomial model has higher variance in the residual variance across genes, on a per-gene basis, the residual variance for the regularized negative binomial model is lower than the zero-inflated negative binomial model and the negative binomial model for the vast majority (86.67%) of genes.

**Sensitivity, specificity, and accuracy.**   We simulated total counts and unique counts measures of 4000 genes separately for both control and condition in 20 sets with varying parameters (a schematic of the simulation model is shown in S2 Fig). For the total count measure, gene sets were simulated in the following way: 4000 genes of control are chosen similarly for all 20 sets, i.e., from a negative binomial distribution with $\mu = 1000$ and $\theta = 100$. Genes that are unaffected by the condition ($n = 3900$) are chosen similarly for all 20 sets, i.e., from a negative binomial distribution with $\mu = 1000$ and $\theta = 100$. The remaining $n = 100$ genes that are truly affected by the condition are chosen from negative binomial distribution with $\mu = (1, 50\ldots, 950)$ and $\theta = 100$, respectively. The intention behind simulating the "true effect" genes with different mean values was to test the model's sensitivity. For the unique counts, transposon insertion values were simulated in the following way: 4000 genes of control are chosen similarly for all 20 sets, i.e., from a negative binomial distribution with $\mu = 20$ and $\theta = 100$. Genes that are unaffected by the condition ($n = 3900$) are chosen similarly for all 20 sets, i.e., from a negative

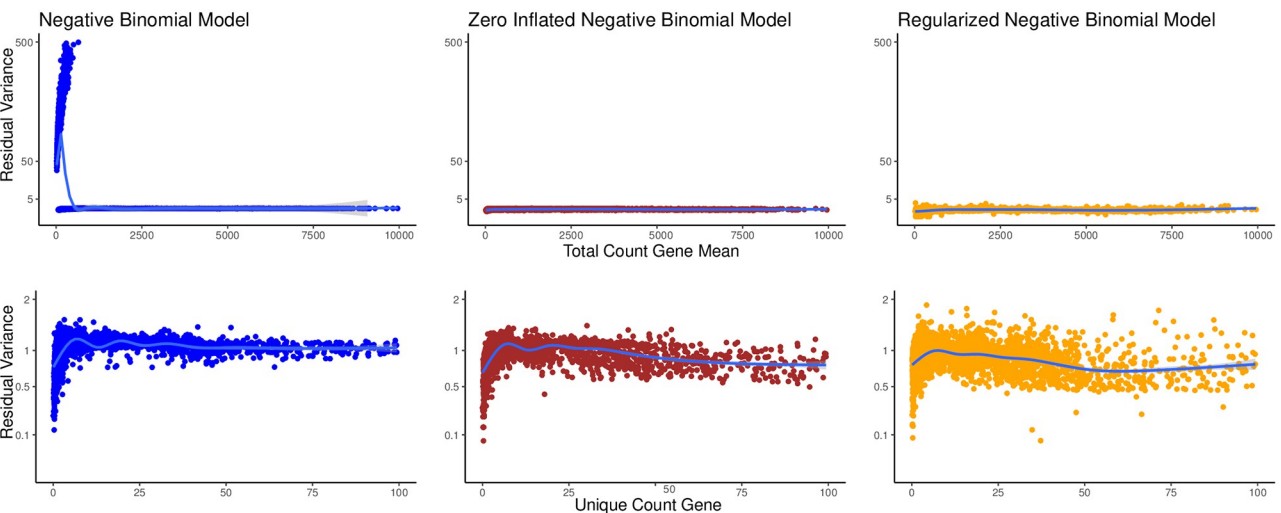

**Fig 3. A simple negative binomial model (left) does not fully capture the variance in genes with low counts.** Zero-inflated negative binomial (center) model overfits count data, attributing almost all variation to strain and conditional effects. As a result, almost every gene exhibits low residual variance. A regularized negative binomial error model (right) successfully captured the mean-variance relationship inherent in the data independent of gene counts. Mean-variance trendline shown in blue for each panel. The results shown are aggregated across multiple simulated conditions described.

binomial distribution with $\mu = 20$ and $\theta = 100$. The remaining $n = 100$ genes that are truly affected by the condition are chosen from negative binomial distribution with $\mu = (1, 2. . ., 20)$ and $\theta = 100$, respectively. This set of simulation data provides true positive genes and true negative genes for the sensitivity/specificity analysis.

The sensitivity, specificity, and accuracy analysis of detected genes was performed for the unregularized negative binomial model, zero-inflated negative binomial model, and regularized negative binomial models. For total counts, regularized negative binomial outperforms the other two models in sensitivity and specificity. For unique counts, regularized negative binomial model has higher sensitivity but slightly lower specificity when the condition mean becomes closer to the control mean. This might be due to fact that regularization of similar effects may yield few false positives for unique counts. In general, sensitivity and accuracy drop as the condition mean becomes closer to the control mean for all models. While taking an intersection of total and unique counts reduces the false positives (higher specificity), it comes at the expense of false negatives (lower sensitivity) especially as the condition mean becomes closer to the control mean. These trends are shown in the Fig 4.

## Analysis of RB-TnSeq data

We fit the regularized negative binomial model to RB-TnSeq data [8]. We selected all *Caulobacter crescentus* and *Pseudomonas fluorescens* experiments and grouped the conditions into carbon-source, nitrogen-source, and stress conditions. The stress conditions, such as heat-stress, antibiotic addition, etc, were conducted in rich media (PYE or LB), while the carbon and nitrogen source changes were conducted in minimal media. The control (wild-type, no stress) experiments were conducted in rich media. The lack of replicate experiments in this data set prevents us from inferring high-confidence conditionally beneficial or conditionally detrimental genes in finer resolution conditions.

**Caulobacter crescentus results.**   For genes with at least one transposon insertion, we identified 3/38/0 (total/unique/overlap) as conditionally essential in carbon, none in nitrogen,

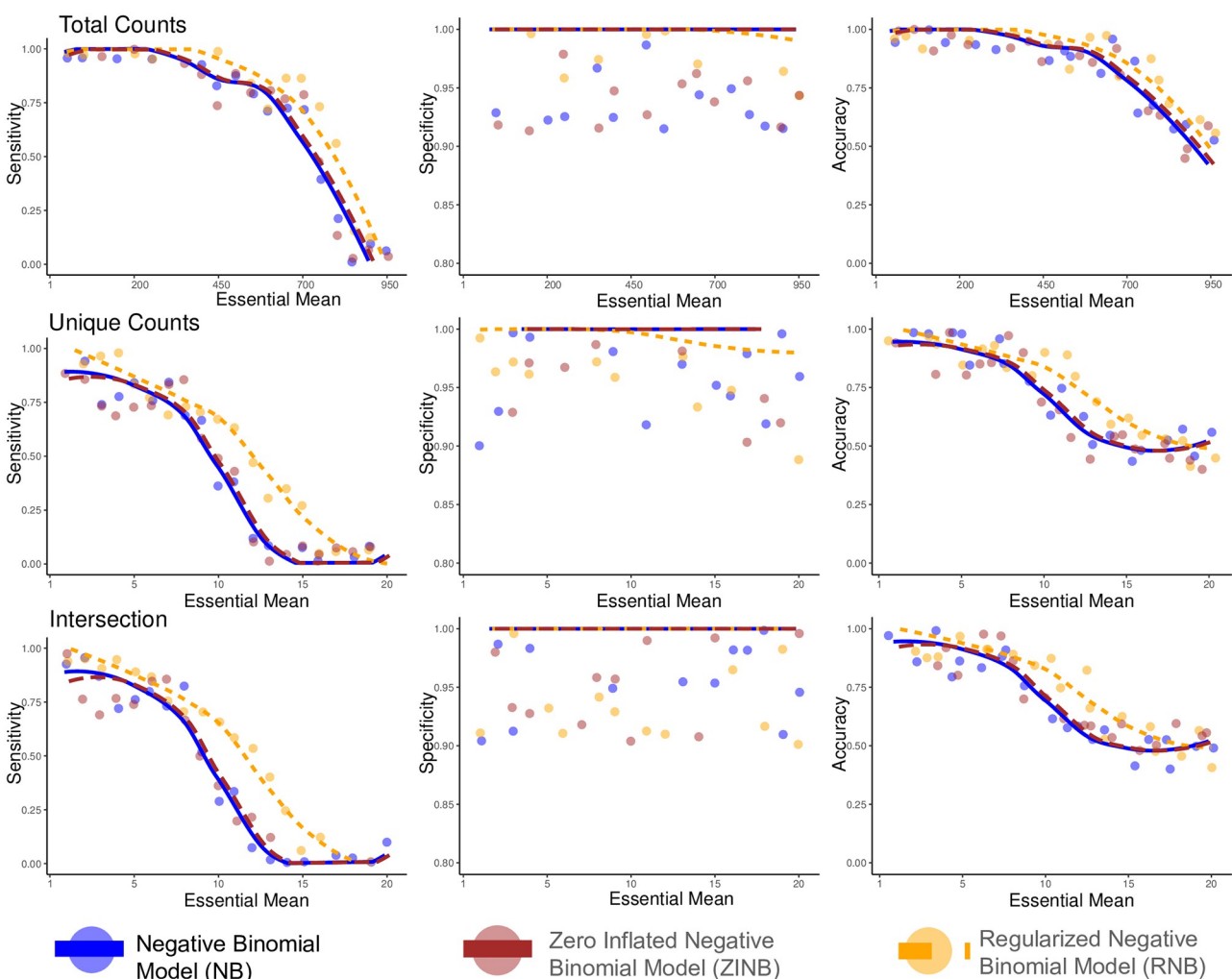

**Fig 4. Comparison of sensitivity, specificity and accuracy of total counts, unique counts and intersection for the simple Negative Binomial model (NB), Regularized Negative Binomial model (RNB), and Zero-inflated negative binomial (ZINB).** The figure shows each model's variation trend of sensitivity, specificity, and accuracy to changes in condition mean for total counts, unique counts and intersection.

and 0/42/0 (total/unique/overlap) in stress as conditionally essential by the criteria described previously. Fig 5(A)–5(C) shows the conditionally beneficial and conditionally detrimental genes in each of the conditions considered for this data set (excluding the conditionally essential genes). Each data point is a gene and genes labeled as red or blue diamonds are called conditionally beneficial or conditionally detrimental by the local FDR criterion. The intersection of blue and red diamonds shows genes that are conditionally beneficial or conditionally detrimental by measures of total insertions and unique insertions. They capture differential fitness for both gene-wise insertions akin to growth and site-specific variation, giving much more confidence in functionally annotating a particular genetic component. It is clear that many genes are called conditionally beneficial (decrease in both total and unique transposon insertions) in both the carbon and nitrogen shift conditions. Our hypothesis is that these genes are required for general biosynthetic processes necessary to survive in minimal media conditions. S1 Table lists all the essential, conditionally essential, conditionally beneficial, and conditionally detrimental genes found in the RB-TnSeq data of *Caulobacter crescentus* by measures of

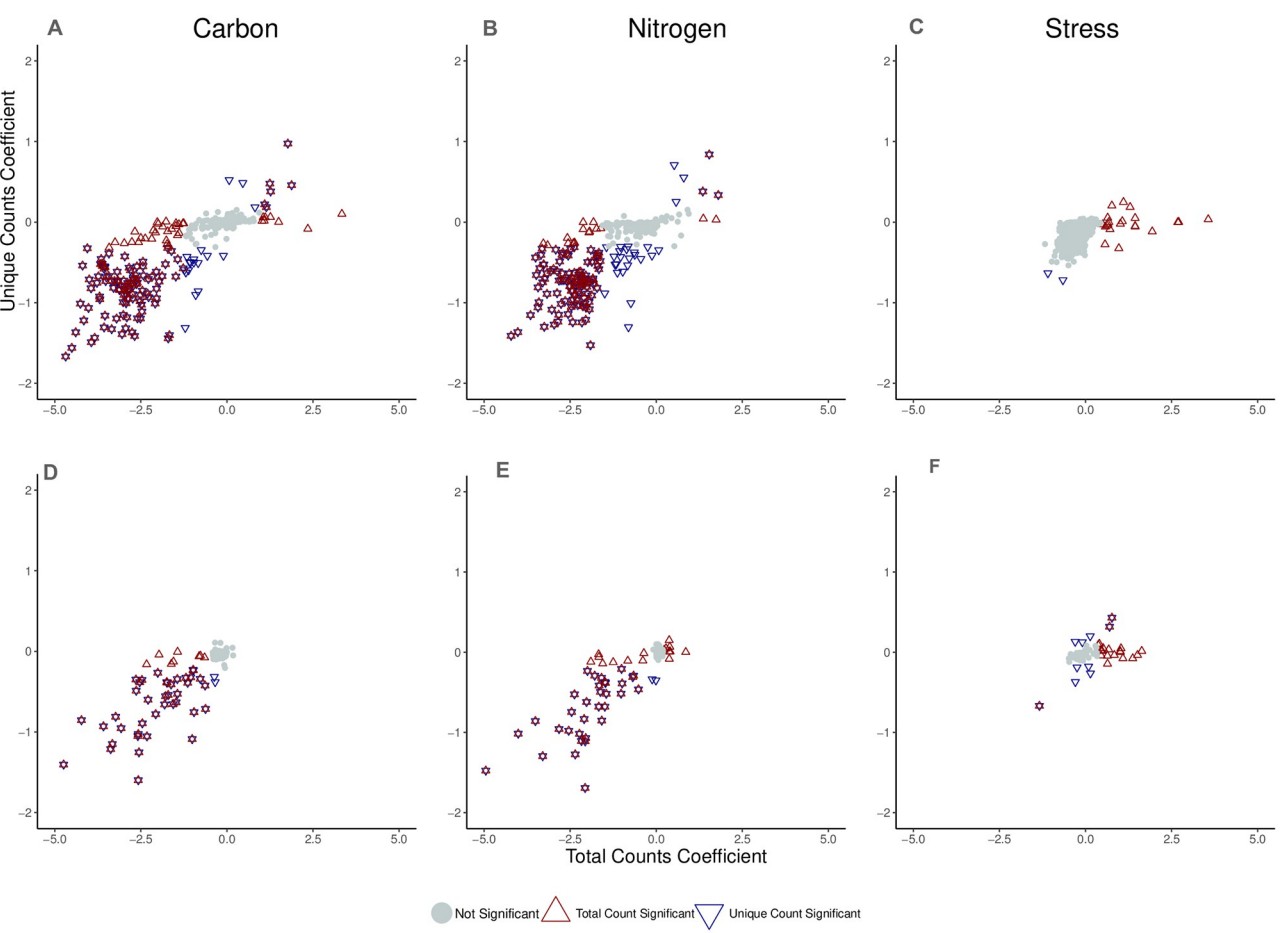

**Fig 5. Conditionally beneficial and conditionally detrimental genes in the published RB-TnSeq data set for *Caulobacter crescentus NA1000* (A-C) and *Pseudomonas fluorescens FW300-N1B4* (D-F) [8].**

the total and unique counts. Fig 6(A) shows the intersection of the gene sets identified in these two conditions and the high degree of overlap and the identities of the genes supports this hypothesis.

In total there are 21 conditionally beneficial or conditionally detrimental genes by total insertion counts and 2 conditionally beneficial or conditionally detrimental genes by unique insertion counts for the stress condition. The two genes that are conditionally beneficial by unique counts: CCNA_03859 (*cenR*), known to be critical for envelope maintenance [33], and CCNA_03346 *ruvC*, a nuclease important for homologous recombination. Because so many of the tested stresses involve the cell envelope either directly (ethanol, polymyxin, etc) or indirectly rely on components in the cell envelope (drug transporters), it is not surprising that a cell envelope maintenance gene like *cenR* would be important for many of these stresses. Because many stresses also lead to DNA damage (cisplatin, metals, etc) we reason that the conditional beneficial nature of *ruvC* stems from its crucial role in resolving crossover junctions, a critical step for DNA damage repair by homologous recombination [34].

***Pseudomonas fluorescens* results.** For genes with at least one transposon insertion, we identified 78/0/0 (total/unique/overlap) as conditionally essential in carbon, none in nitrogen, and 89/0/0 (total/unique/overlap) in stress as conditionally essential by the criteria described previously. S2 Table lists all the essential, conditionally essential, conditionally beneficial, and

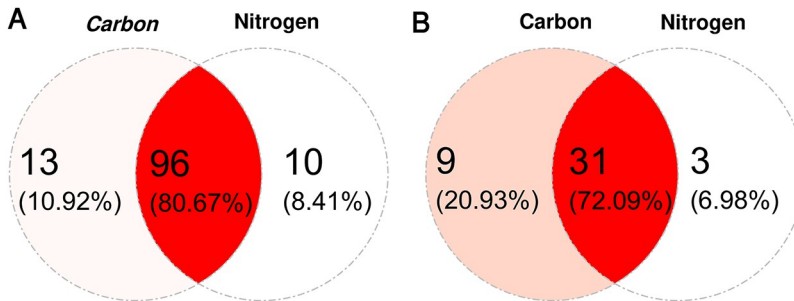

**Fig 6. Venn diagram showing a high degree of overlap between genes identified in carbon and nitrogen shift conditions in *Caulobacter crescentus NA1000* (A) and *Pseudomonas fluorescens FW300-N1B4* (B) indicating genes involved in the shift to minimal media are identified.**

conditionally detrimental genes found in the RB-TnSeq data of *Caulobacter crescentus* by measures of the total and unique counts. Fig 5(D) and 5(E) shows the conditionally beneficial and detrimental as identified by the mode in each of the conditions considered for this data set. There are two conditionally detrimental genes and one conditionally beneficial gene by both measures of total insertion counts and unique insertion counts for the stress condition. First, a conditionally detrimental gene, Pf1N1B4_2858 (*CbrB*), is a two-component sensor needed for cells to use a variety of carbon or nitrogen sources [35]. The second conditionally detrimental gene is Pf1N1B4_1906, a Shikimate 5-dehydrogenase which would influence synthesis of aromatic amino acids. The conditionally beneficial gene is Pf1N1B4_2106, also known as OxyR, a hydrogen peroxide-inducible transcriptional activator which controls expression of oxidative stress response proteins. The conditional beneficiality of this gene makes mechanistic sense because many of the stress conditions impact the oxidative stress system.

## Analysis of Tn-seq data

We fit the regularized negative binomial model to our own Tn-seq data on *Caulobacter crescentus* experiments described previously. Of the 4,084 genes with at least one transposon insertion, we identified 84/109/64 (total/unique/overlap) as conditionally essential in heat shock, 23/22/13 (total/unique/overlap) in canavanine, 237/251/210 (total/unique/overlap) in Δ*lon*, 211/233/188 (total/unique/overlap) in Δ*lon*+heat shock, and 253/286/231 (total/unique/overlap) in Δ*lon*+canavanine by the criteria described previously. We identified 26/53/17 (total/unique/overlap) as conditionally beneficial or conditionally detrimental in heat shock, 20/22/10 (total/unique/overlap) in canavanine, 132/2/2(total/unique/overlap) in Δ*lon*, 13/19/9 (total/unique/overlap) in Δ*lon*+heat shock, and 24/34/14(total/unique/overlap) in Δ*lon*+canavanine by our modeling framework. S3 Table lists all the essential, conditionally essential, significant conditionally beneficial, and significant conditionally detrimental genes found in the Tn-seq data by measures of the total and unique counts. Fig 7 shows the conditionally beneficial and conditionally detrimental genes in each of the conditions considered for this data set (excluding the conditionally essential genes). Each data point is a gene and genes labeled as triangles are called conditionally beneficial or conditionally detrimental by the local FDR criterion.

**Conditionally beneficial genes.** In *wild-type* strains under heat stress conditions, we found six genes that are conditionally beneficial. One of these (*metK*) is a known substrate of the chaperone GroEL [36], suggesting that during heat stress prolific misfolding of MetK could result in a higher need for the metK gene in *Caulobacter*. Under canavanine conditions, there is only one gene found beneficial in this condition, the *katG* gene, a peroxidase-catalase

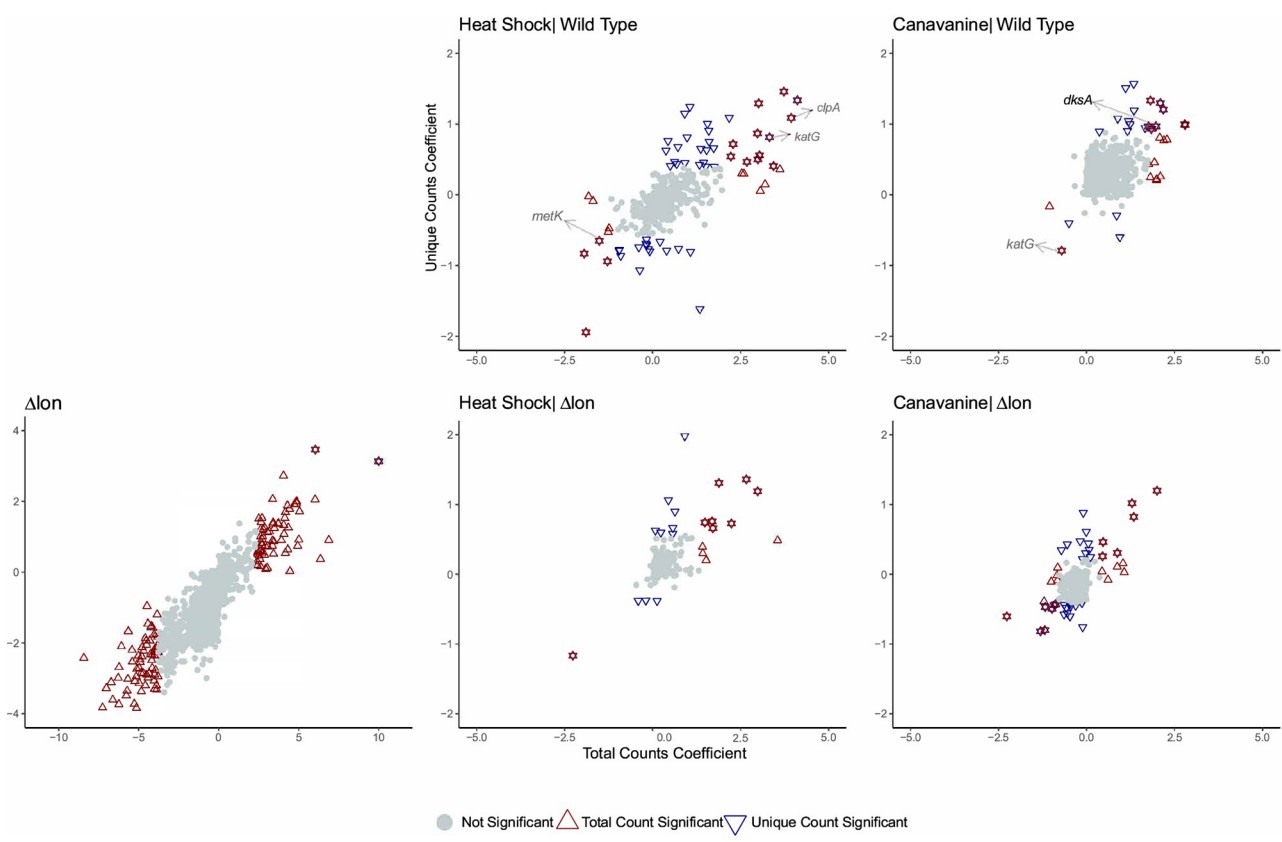

**Fig 7. Analysis of transposon sequencing data of *Caulobacter crescentus*.** Shown are the regularized model coefficients for the genetic background effect for Δ*lon* and the nested environmental conditions: heat shock and canavanine for total count and unique count data.

gene that is critical for oxidative tolerance in stationary phase [37]. Neither of these genes seem to be conditionally beneficial during stress conditions for cells lacking the Lon protease, suggesting that these mutant strains respond differently to protein homeostasis stresses.

**Conditionally detrimental genes.** We found twelve genes that were conditionally detrimental during heat stress and eight during canavanine stress in wildtype strains. For those detrimental during heat stress, we were intrigued to find *katG* as well, suggesting that while *katG* is important for tolerating canavanine induced protein misfolding, its presence confers less fitness when cells are subject to heat stress. We note that during canavanine stress, the *dksA* gene becomes detrimental. *dksA* was identified as a multicopy suppressor of growth defects stemming from loss of the DnaK chaperone and it is known to inhibit ribosome synthesis [38], suggesting a strong role in proteostasis. We speculate that loss of *dksA* may guard against protein misfolding stress resulting from canavanine misincorporation, or improve ribosome capacity which is taxed due to misincorporation of canavanine. Again, while we see similar numbers of genes being conditionally detrimental in cells lacking Lon under these stress conditions, there is no overlap in the sets, suggesting a different program in place for stress response.

**Validation experiments.** Our model has identified *clpA* to be conditionally detrimental by both measures of total and unique counts in the *wt* background under heat stress. To validate this we performed competitive mutant fitness assays comparing the growth rate of *wt* and Δ*clpA* in competition with a *wt* strain constitutively expressing the fluorescent reporter, Venus. The competition assay results in Fig 8 shows that heat-stress (42°C) compensates for

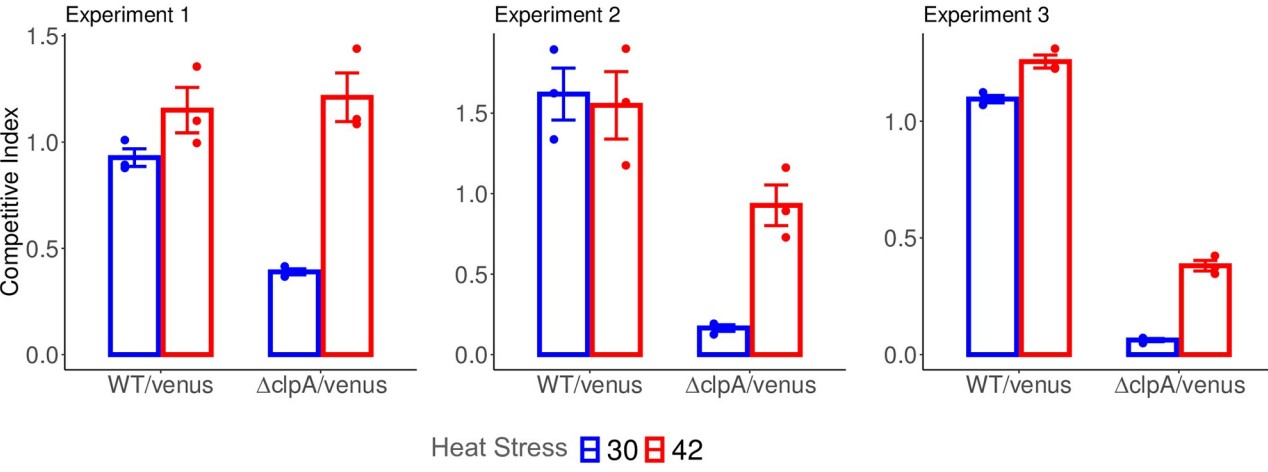

**Fig 8. Competitive mutant fitness experiment comparing fitness of wild-type and ΔclpA under heat stress.** Y-axis corresponds to the ratio of cells after 24 hours growth either with no heat stress (30) or after a transient 42 degrees C heat stress (42) compared to the Venus reporter strain. Ratios of initial mixtures normalized to 1. Error bars indicate the standard error of the mean of each group.

the fitness defect caused by the loss of *clpA* under normal conditions (30˚C) across three biological replicates.

## Conclusion

We have presented a model-based method that uses regularized negative binomial regression to estimate the change in transposon insertions attributable to gene-environment changes without transformations or uniform normalization. Simulation experiments indicate that the regularized negative binomial model performs well without over-fitting. When applied to RB-TnSeq and Tn-Seq using both total and unique data, the model is able to identify sets of conditionally beneficial and conditionally detrimental genes for each perturbation that shed light on their functions and roles during various stress conditions.

## Supporting information

**S1 Table. RB-TnSeq data analysis of *Caulobacter crescentus*.** List of essential, conditionally essential, conditionally beneficial, and conditionally detrimental genes found in the RB-TnSeq data of *Caulobacter crescentus* by measures of the total and unique counts.
(XLSX)

**S2 Table. RB-TnSeq data analysis of *Pseudomonas fluorescens*.** List of essential, conditionally essential, conditionally beneficial, and conditionally detrimental genes found in the RB-TnSeq data of *Pseudomonas fluorescens* by measures of the total and unique counts.
(XLSX)

**S3 Table. Tn-seq data analysis results.** List of essential, conditionally essential, significant conditionally beneficial, and significant conditionally detrimental genes found in the Tn-seq data by measures of the total and unique count, total counts.
(XLSX)

**S1 Fig. Comparison of regularized negative binomial model (right) with a simple negative binomial model (left) and Zero-inflated negative binomial (center).** These results show data

for only the control and one condition in contrast to the multiple condition results shown in Fig 3.
(TIF)

**S2 Fig. Schematic representation of our simulation framework to test sensitivity, specificity, and accuracy of our model.** A total of 20 sets each containing 4000 genes in a control condition setup are simulated for total counts and unique counts separately according to schema and parameters shown in the figure.
(TIF)

## Author Contributions

**Conceptualization:** Patrick Flaherty.

**Data curation:** Berent Aldikacti, Rilee Zeinert.

**Formal analysis:** Vishal Sarsani, Shai He.

**Funding acquisition:** Peter Chien, Patrick Flaherty.

**Investigation:** Vishal Sarsani, Peter Chien, Patrick Flaherty.

**Methodology:** Vishal Sarsani, Shai He, Patrick Flaherty.

**Project administration:** Patrick Flaherty.

**Resources:** Peter Chien.

**Software:** Vishal Sarsani, Shai He.

**Supervision:** Peter Chien, Patrick Flaherty.

**Validation:** Berent Aldikacti.

**Visualization:** Vishal Sarsani, Berent Aldikacti.

**Writing – original draft:** Vishal Sarsani, Berent Aldikacti, Peter Chien, Patrick Flaherty.

**Writing – review & editing:** Vishal Sarsani, Berent Aldikacti, Peter Chien, Patrick Flaherty.

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
