## [Decision Letter · Decision Letter 0]

8 Oct 2021

Dear Dr Flaherty,

Thank you very much for submitting your manuscript "Model-based identification of conditionally-essential genes from transposon-insertion sequencing data" for consideration at PLOS Computational Biology.

As with all papers reviewed by the journal, your manuscript was reviewed by members of the editorial board and by several independent reviewers. In light of the reviews (below this email), we would like to invite the resubmission of a significantly-revised version that takes into account the reviewers' comments.

We cannot make any decision about publication until we have seen the revised manuscript and your response to the reviewers' comments. Your revised manuscript is also likely to be sent to reviewers for further evaluation.

Sincerely,

Nicola Segata

Associate Editor

PLOS Computational Biology

Sushmita Roy

Deputy Editor

PLOS Computational Biology

Reviewer's Responses to Questions

**Comments to the Authors:**

Reviewer #1: In this manuscript, Sarsani and colleagues develop a negative binomial regression model for the analysis of conditional and genetic interaction experiments based on the transposon insertion sequencing technology. The major innovations over previous methods are in the regularization through the use of priors on coefficients, and independently testing both unique and total counts with a method for joint FDR control.

After reviewing the manuscript and previous reviews, I believe the work is fundamentally sound. As it has already been through one round of review, I will restrict myself to comments that could easily be addressed through text changes.

One comment to make the manuscript more accessible to its target audience: the gene X environment experimental paradigm the authors investigate is generally referred to as a genetic interaction study – referencing this term in the abstract and manuscript text would increase the ability of those conducting these sorts of experiments to find the method. Additionally, it would probably be worth contrasting the authors’ proposed method with DeJesus et al 2017 NAR “Statistical analysis of genetic interactions in Tn-Seq data”.

A second comment on terminology: the authors use the terms ‘conditionally essential’ and ‘conditionally dispensable’ to refer to genes whose deletion have negative and positive effects on fitness, respectively. The term ‘essential’ strictly refers to genes whose absence completely abrogates growth in pure culture (not just a fitness defect), though this term is frequently abused in the pooled screening literature so perhaps not much point in fighting the tide. On the other hand, ‘dispensable’ just means non-essential – I think what the authors mean is something more like ‘conditionally detrimental’, as all genes with no effect on fitness either way under the assay conditions are clearly dispensable, but probably not especially interesting (at least in the studied condition).

A final point that could use some clarification is the theoretical basis behind using both total counts and unique counts. I could see an argument based on PCR bias or jackpotting – I don’t think this needs to be proven, but it would be nice to have some speculation or intuition on why this is a sensible thing to do besides that it reduces the total number of discoveries. It may also be worth discussing the patterns seen in figure 3 in this context – do the authors have any speculation as to what is happening with the blue and red triangles?

Minor point: on page 6, lines 206 and 209, I’m not sure what delta YFG means (besides a gene deletion background by context) – it would be worth either clarifying what this acronym means, or using something that more obviously just a placeholder (e.g. delta gene). Similarly, on the same page, the authors abruptly switch to talking about delta lon with no context – this makes sense later in the manuscript but is a bit confusing at that point.

Reviewer #2: This manuscript from Sarsani et al. present a linear modeling approach to the

modeling transposon sequencing data. The methods are (by now) fairly classical

ones, but they are apriori appropriate. The benchmarking is suggestive, but not

conclusive.

MAJOR COMMENTS

1. Lines 280-1: Naturally taking the intersection will reduce the false

positives, but it will also increase the false negatives. In fact, while the

authors here mention this intersection, this is not used at all later as the

counts for total/unique are presented seperately (see #6 below). The issue of

false negatives (or alternative, of power or precision/recall tradeoffs) is

never discussed.

2. The results of the simulation experiment are only indirect validation as

they do not directly address the question of whether the model correctly picks

out the ground truth set of essential/dispensable genes, which is the task that

the authors are trying to solve.

3. Lines 100-8: I find this formulation (used through the manuscript), to be

fairly confusing. G is the set of genes, but also the set of genetic

backgrounds (which is true for a specific type of background, single-gene

knockouts). Doesn't R depend on g & e?

Finally, the constant reference to "essential/dispensable" is confusing. It

would seem from the preceding text that it should read "essential" only

(although, later we see that deviations can be in both directions, which is

what I think motivates this presentation).

MINOR COMMENTS

4. Line 59: probably a better phrasing would be "robust to differences in

transposon sequencing technology".

5. Fig 4: Can the authors add the raw number of genes (not just percentages) to

the diagrams?

6. Lines 370-80: Can the authors add the overlap to their gene counts? For

example, when they present "337/395" for the heat shock case, how many genes

are in the overlap between total and unique?

**Have the authors made all data and (if applicable) computational code underlying the findings in their manuscript fully available?**

Reviewer #1: Yes

Reviewer #2: Yes

PLOS authors have the option to publish the peer review history of their article (what does this mean?). If published, this will include your full peer review and any attached files.

Reviewer #1: No

Reviewer #2: No
---

## [Decision Letter · Decision Letter 1]

15 Jan 2022

Dear Dr Flaherty,

Thank you very much for submitting your manuscript "Model-based identification of conditionally-essential genes from transposon-insertion sequencing data" for consideration at PLOS Computational Biology. As with all papers reviewed by the journal, your manuscript was reviewed by members of the editorial board and by several independent reviewers. The reviewers appreciated the attention to an important topic. Based on the reviews, we are likely to accept this manuscript for publication, providing that you modify the manuscript according to the review recommendations.

Sincerely,

Nicola Segata

Associate Editor

PLOS Computational Biology

Sushmita Roy

Deputy Editor

PLOS Computational Biology

[LINK]

Reviewer's Responses to Questions

**Comments to the Authors:**

Reviewer #1: The authors have responded thoughtfully to my previous comments, and I have no further comments or concerns.

Reviewer #2: I thank the authors for their improvements and the manuscript is significantly

better. I have no further scientific comments, but I still have some remarks

with respect to presentation.

The newly added "Sensitivity, Specificity, and Accuracy" indeed does address

the comment I raised in the previous round, but it is very hard to follow. The

description should be rewritten to be clearer or a cartoon representation of

the simulated settings could be added for clarity.

The supplementary figures are misnumbered. I think both the flowchart (referred

to as SFig 1 in the text) and the results of the sensitivity experiments

(referred to as SFig 3) could be moved to the main text.

Generally speaking, the figures would benefit from some minor polish for

readability: the fonts are too small, sometimes abbreviations are used when

there appears to be enough room to spell out the terms, some of the alignments

seem off (particularly in supplementary figure 3).

**Have the authors made all data and (if applicable) computational code underlying the findings in their manuscript fully available?**

Reviewer #1: Yes

Reviewer #2: Yes

PLOS authors have the option to publish the peer review history of their article (what does this mean?). If published, this will include your full peer review and any attached files.

Reviewer #1: No

Reviewer #2: No

Figure Files:

Data Requirements:

Reproducibility:

References:

---

## [Editor Report · Decision Letter 2]

9 Feb 2022

Dear Dr Flaherty,

We are pleased to inform you that your manuscript 'Model-based identification of conditionally-essential genes from transposon-insertion sequencing data' has been provisionally accepted for publication in PLOS Computational Biology.

Best regards,

Nicola Segata

Associate Editor

PLOS Computational Biology

Sushmita Roy

Deputy Editor

PLOS Computational Biology

---

## [Editor Report · Acceptance letter]

1 Mar 2022

PCOMPBIOL-D-21-01215R2 

Model-based identification of conditionally-essential genes from transposon-insertion sequencing data

Dear Dr Flaherty,

I am pleased to inform you that your manuscript has been formally accepted for publication in PLOS Computational Biology. Your manuscript is now with our production department and you will be notified of the publication date in due course.

With kind regards,

Livia Horvath
